# Early-Life Metabolic and Hormonal Markers in Blood and Growth until Age 2 Years: Results from a Randomized Controlled Trial in Healthy Infants Fed a Modified Low-Protein Infant Formula

**DOI:** 10.3390/nu13041159

**Published:** 2021-04-01

**Authors:** Stefanie M. P. Kouwenhoven, Manja Fleddermann, Martijn J. J. Finken, Jos W. R. Twisk, Eline M. van der Beek, Marieke Abrahamse-Berkeveld, Bert J. M. van de Heijning, Dewi van Harskamp, Johannes B. van Goudoever, Berthold V. Koletzko

**Affiliations:** 1Emma Children’s Hospital, Vrije Universiteit, University of Amsterdam, Amsterdam UMC, 1081 Amsterdam, The Netherlands; s.kouwenhoven@amsterdamumc.nl; 2Department of Peadiatrics, Dr. von Hauner Children’s Hospital, LMU University Hospitals, LMU—Ludwig-Maximilians-Universität Munich, 80337 Munich, Germany; manja-fleddermann@web.de (M.F.); berthold.koletzko@med.uni-muenchen.de (B.V.K.); 3Department of Paediatric Endocrinology, Emma Children’s Hospital, Vrije Universiteit, Amsterdam UMC, 1081 Amsterdam, The Netherlands; m.finken@amsterdamumc.nl; 4Epidemiology and Data Science, Amsterdam UMC, 1081 Amsterdam, The Netherlands; jwr.twisk@amsterdamumc.nl; 5Department of Pediatrics, University Medical Center Groningen, University of Groningen, 9713 Groningen, The Netherlands; e.m.van.der.beek@umcg.nl; 6Danone Nutricia Research, 3584 Utrecht, The Netherlands; marieke.abrahamse@danone.com (M.A.-B.); bert.vandeheijning@danone.com (B.J.M.v.d.H.); 7Stable Isotope Laboratory, University of Amsterdam, Amsterdam UMC, 1105 Amsterdam, The Netherlands; d.vanharskamp@amsterdamumc.nl

**Keywords:** early childhood, infant nutrition, growth, body composition, childhood obesity, protein intake

## Abstract

Background: High protein intake in early life is associated with an increased risk of childhood obesity. Dietary protein intake may be a key mechanistic modulator through alterations in endocrine and metabolic responses. Objective: We aimed to determine the impact of different protein intake of infants on blood metabolic and hormonal markers at the age of four months. We further aimed to investigate the association between these markers and anthropometric parameters and body composition until the age of two years. Design: Term infants received a modified low-protein formula (mLP) (1.7 g protein/100 kcal) or a specifically designed control formula (CTRL) (2.1 g protein/100 kcal) until 6 months of age in a double blinded RCT. The outcomes were compared with a breast-fed (BF) group. Glucose, insulin, leptin, IGF-1, IGF-BP1, -BP2, and -BP3 levels were measured at the age of 4 months. Anthropometric parameters and body composition were assessed until the age of 2 years. Groups were compared using linear regression analysis. Results: No significant differences were observed in any of the blood parameters between the formula groups (*n* = 53 mLP; *n* = 44 CTRL) despite a significant difference in protein intake. Insulin and HOMA-IR were higher in both formula groups compared to the BF group (*n* = 36) (*p* < 0.001). IGF-BP1 was lower in both formula groups compared to the BF group (*p* < 0.01). We found a lower IGF-BP2 level in the CTRL group compared to the BF group (*p* < 0.01) and a higher IGF-BP3 level in the mLP group compared to the BF group (*p* = 0.03). There were no significant differences in glucose, leptin, and IGF-1 between the three feeding groups. We found specific associations of all early-life metabolic and hormonal blood parameters with long-term growth and body composition except for IGF-1. Conclusions: Reducing protein intake by 20% did not result in a different metabolic profile in formula-fed infants at 4 months of age. Formula-fed infants had a lower insulin sensitivity compared to breast-fed infants. We found associations between all metabolic and hormonal markers (except for IGF-1) determined at age 4 months and growth and body composition up to two years of age.

## 1. Introduction

There is evidence to suggest that protein intake during the first months of life has long-lasting effects on growth, body composition, and obesity risk [1,2,3,4,5]. The mechanisms through which an increased protein intake may affect growth and body composition remain to be clarified. The anabolic hormones insulin and insulin-like growth factor-1 (IGF-1) are responsive to fluctuations in protein intake [6] and have growth-stimulating properties [7,8].

Formula-fed infants have been shown to have higher concentrations of insulin [9,10,11] and IGF-1 [11,12,13,14] compared to breast-fed infants. This has often been attributed to the higher protein contents of formula feeding. Indeed, a randomized controlled trial investigating the effect of formula feeding with different protein content found that concentrations of IGF-1 and C-peptide excretion (which is an indicator of insulin secretion) were higher among infants receiving formula with a higher protein content [11].

The bioactivity of IGF-1 is regulated by six IGF-binding proteins (IGF-BP1–6). Circulating IGF-1 is primarily bound to IGF-BP3 in addition to an acid-labile subunit, thus offering protection from early degradation in the circulation [15]. In target tissues, liberated IGF-1 can bind to the IGF-1 receptor [16]. Some IGF-BPs possess intrinsic biological activities through IGF-independent mechanisms [17,18]. Accumulating evidence suggests that IGF-BP1 and IGF-BP2 may increase insulin sensitivity directly [18] and cluster with components of the metabolic syndrome in obese children [19] Hence, these IGF-BPs may serve as a useful biomarker of insulin sensitivity in early childhood [20].

The appetite-regulating hormone leptin may also explain the link between protein intake and infant growth and body composition. Leptin is involved in regulating satiety and energy balance [21]. Circulating leptin levels have been shown to correlate well with the total fat mass [22]. Protein intake may influence the amount of adipose tissue and, therefore, can circulate leptin.

We found that feeding an infant formula with a modified amino acid profile and a lower protein content (mLP formula) is safe and supports adequate growth despite a lower urea concentration similar to infants receiving an infant formula with a standard amount of protein (CTRL formula) [23]. Furthermore, no differences in growth parameters or body composition were observed between the mLP group and the CTRL group at the ages of 4 and 6 months. Since the infants in the mLP group had a lower total protein intake, we hypothesized that infants fed a mLP formula have a different metabolic profile than infants fed CTRL formula more closely resembling that of breast-fed infants. We further investigated the associations between metabolic/hormonal markers and anthropometric parameters, including body composition.

## 2. Materials and Methods

This study was part of a double-blind randomized controlled trial (ProtEUs study) conducted in two centers: Amsterdam UMC (location VU University Medical Center, Amsterdam, the Netherlands) and Dr. von Hauner Children’s Hospital (LMU—Ludwig-Maximilians-Universität, Munich, Germany). Healthy, term-born, formula-fed (*n* = 178), and breast-fed (*n* = 67) infants were enrolled between 22 October 2014 and 29 December 2016. Formula-fed infants eligible for participation were randomized to receive either the mLP with a protein content of 1.7 g/100 kcal or CTRL formula with a protein content of 2.1 g/100 kcal from baseline (an average age of one month) up to the age of 6 months. All infants were fed ad libitum throughout the study period. The study formulas were isoenergetic (67 kcal/100 mL) and had identical nutritional composition except for protein and lactose; the latter was used to make the formula isoenergetic. Volume intake in both formula groups was similar across all ages, and no compensatory volume intake was observed in the mLP group [23]. This confirms that the targeted difference of 20% in protein intake was established between the mLP and CTRL groups during the intervention period.

The outcomes were compared with a breast-fed (BF) group. The breast-fed group consisted of infants of parents who intended to (fully) breast-feed for 6 months. Parents were allowed to introduce complementary feeding no earlier than after the age of 17 weeks in line with national guidelines. The trial was approved by the Institutional Review Boards of the VU University Medical Center Amsterdam and the Medical Faculty, LMU Munich. The study was conducted according to International Conference on Harmonisation–Good Clinical Practice (ICH-GCP) principles and in compliance with the principles of the Declaration of Helsinki. An independent data safety monitoring board assessed for trial safety. Written informed consent was obtained from all participants’ parents or guardians. Further details of the study are described elsewhere [23].

### 2.1. Procedures

The intervention period started at an average age of one month and lasted until the age of 6 months. Infants visited the hospital 3 times during the intervention period: at an average age of 1 month (baseline), at 4 months of age, and at 6 months of age. The follow-up period included study visits at 1 and 2 years of age. During these visits, anthropometry and body composition measurements were performed by trained study personnel according to standard protocols. At the age of 4 months, a venous blood sample from a hand vein was obtained after at least 3 h of fasting. Parents were allowed to introduce complementary feeding no earlier than after the age of 4 months in line with national guidelines.

### 2.2. Anthropometrics

Weight was measured (without clothing or diaper) to 0.5-g accuracy on a balance scale (MARSDEN, Rotherham, UK). At the age of 1 year, length was measured with a flexible measuring board (SECA, Birmingham, UK) to the nearest 0.1 cm. At 2 years, height was measured with a digital stadiometer (SECA, Birmingham, UK) to the nearest 0.1 cm. The average of two measurements was used in the analysis.

### 2.3. Body Composition

Air-displacement plethysmography (ADP) technique is conducted as follows: At the age of 1, 4, and 6 months, we used the Pea Pod (Cosmed, Concord, MA, USA) and at the age of 2 years we used the Bod Pod (Pediatric Option Body Composition System; Cosmed, Concord, MA, USA) to measure body composition.

### 2.4. Deuterium Oxide Dilution (D_2_O) Technique

At the age of 1 year, body composition was estimated using the deuterium dilution technique [24,25]. After the collection of a baseline saliva sample (t0), a small amount of 0.2 mL or 0.4 mL D_2_O per kilogram body weight was orally administered. Saliva was collected using a saliva collection kit at 5 distinct time points after the administration of D_2_O, (in hours: t6 or t10, t24, t48, t96, and t168). All samples were stored frozen at −20 °C prior to analysis. The saliva was recovered from the kits by centrifugation (10 min, 4000× *g*, 20 °C). The details of the procedures are described elsewhere [26]. A fat-free mass was calculated from total body water using age- and sex-specific conversion factors for children up to 2 years of age [27].

### 2.5. Laboratory Analyses

Venous blood was collected in a 2.5-mL serum tube and a 0.5-mL heparin tube. This was divided into six different aliquots after centrifugation (10 min, 1800× *g*, 20 °C). Samples were stored at −80 °C and thawed only once just before analysis. Glucose levels were assessed on a Beckman Coulter AU5800/AU 680 (enzymatic UV-assay) and insulin concentration on a Roche Cobas 8000/e411 (electrochemiluminescence immunoassay (ECLIA)). Levels of IGF-1, IGF-binding proteins, and leptin were estimated using ELISA (Mediagnost Inc., Reutlingen, Germany).

### 2.6. Outcomes

Differences existed between the concentration of glucose, insulin, leptin, IGF-1, IGF-BP1, IGF-BP2, and IGF-BP3, and the homeostasis model assessment (HOMA-IR) [28] between mLP, CTRL, and BF groups at the age of 4 months.

Associations between these metabolic and hormonal markers and anthropometric parameters and body composition until the age of 2 years (non-randomized).

## 3. Statistical Analyses

Insulin, HOMA-IR, and IGF-BP1 were log transformed (natural log-transformation) before statistical analysis. Normally distributed data were presented as mean ± SD and skewed data as medians (interquartile ranges—IQRs). The differences in outcome between the groups at the age of four months were analyzed with linear regression analysis and adjustments were made for sex. We compared two feeding groups per analysis (mLP vs. CTRL, mLP vs. BF, CTRL vs. BF).

Associations between metabolic and hormonal markers and anthropometric parameters and body composition were analyzed using a linear mixed model analysis. Linear mixed models were used to consider the dependency of the observations within the child. Interactions were added to obtain the associations at different time points (4 months, 6 months, 1 year, and 2 years). The analyses were adjusted for feeding group and sex. Statistical analyses were performed using the Statistical Package for the Social Sciences (SPSS) version 26, and a significance level of 5% was used for all comparisons.

## 4. Results

At the age of 4 months, we collected blood samples of 134 infants out of the 238 infants who participated in this trial at that age (Table 1). Of these samples, 53 were fed with mLP formula, 45 with CTRL formula, and 36 were breast-fed. The anthropometrics and body composition of these children are presented in Table 2, and the metabolic and hormonal markers are shown in Table 3. No blood samples were obtained from 104 infants due to unsuccessful draws or parental objection.

### 4.1. Differences between Feeding Groups

No significant differences in metabolic and hormonal markers were found between the mLP group and the CTRL group (Table 4, Figure 1 and Figure 2). Nor did we find any significant differences in glucose, IGF-1 concentration (Table 4), or leptin levels between the three feeding groups (Table 4, Figure 1). However, both insulin level and HOMA-IR were significantly higher in formula-fed infants compared to breast-fed infants (Table 4, Figure 2) while IGF-BP1 was significantly lower in both formula groups compared to the BF group (Table 4). IGF-BP2 was significantly lower in the CTRL group compared to the BF group but not different between the mLP and the BF group (Table 4).

In contrast, IGF-BP3 was significantly higher in the mLP group compared to the BF group but not different between the BF group and the CTRL group (Table 4). The ratio IGF-1/IGFBP-3 was not different between the feeding groups (data not shown).

### 4.2. Associations between Metabolic and Hormonal Markers and Anthropometric Parameters at 4 Months of Age

At the age of 4 months, we found a positive association between leptin and all anthropometric outcomes, including body composition. We further found that IGF-BP1 was negatively associated with length. IGF-BP2 was negatively associated with weight and fat-free mass while IGF-BP3 was positively associated with weight and fat-free mass (Table 5).

### 4.3. Associations between Early Metabolic and Hormonal Markers and Long-Term Anthropometric Parameters

Glucose at 4 months was positively associated with fat free mass at 2 years. In contrast, insulin and HOMA were negatively associated with fat free mass at that age. No other associations were found between these markers and long-term growth.

Leptin at 4 months was positively associated with all anthropometric outcomes at all ages except for weight and fat free mass at 1 year as well as fat free mass and fat mass percentage at 2 years. IGF-1 at 4 months was not associated with anthropometric outcomes until the age of 2 years. IGF-BP1 and IGF-BP2 at 4 months were negatively associated with weight and length from 6 months up to 2 years. Furthermore, IGF-BP1 was negatively associated with fat free mass at 6 months and 1 year. At the age of 2 years, we found a negative association with fat mass and fat mass percentage. In contrast, IGF-BP2 was positively associated with fat free mass at 6 months. However, this association was negative at the age of 1 year. IGF-BP3 at 4 months was positively associated with weight, fat mass, and fat free mass at the age of 6 months. IGF-BP3 was still positively associated with weight at the age of 1 year. Again a positive association was found with fat free mass at 2 years.

Excluding the body composition results obtained at the age of 1 year with D_2_O, there was no change in the results (level of significance) except for the association between insulin and fat mass percentage (Appendix A). In contrast to the complete analysis (*p* = 0.06, Table 5), we found an association between insulin and fat mass percentage at the age of 2 years in the analysis with exclusive ADP body composition measurements (*p* = 0.04, Appendix A).

## 5. Discussion

In contrast to our hypothesis, we found similar levels of metabolic and hormonal markers along with similar growth [23] in infants fed mLP formula and infants fed CTRL formula at the age of 4 months despite the lower protein intake in the mLP infants.

Previously, two double-blind randomized controlled trials investigated the effect of protein content in infant formula on metabolic hormones in healthy infants. The CHOP study investigated the long-term effects of total protein intakes of a higher protein infant formula (2.9 g protein/100 kcal for 0–4 months and 4.4 for 4–12 months) compared with formulas with lower protein amounts (1.77 and 2.2 g protein/100 kcal, respectively) [11]. The EPOCH study examined the effect of a different protein intake on IGF-1 concentration. Infants consumed a high-protein formula (2.7 g protein/100 kcal) or a low-protein formula (1.8 g protein/100 kcal) until the age of 1 year.

In line with these studies, we found that formula-fed infants were more insulin-resistant as seen by the levels of insulin, IGF-BP1/IGF-BP2, and HOMA-IR than breast-fed infants. However, conflicting results are found upon comparing markers of insulin resistance in formula-fed infants with different protein intakes [10,11]. These different findings might be caused by the difference in total protein intake between the intervention- and the control group included in these studies. In these studies, the formulas used differed by 1.1 (age 0–4 months) and 2.2 g protein/100 kcal (age 4–12 months) (CHOP), as well as 0.9 g protein/100 kcal (EPOCH).

Although short-term associations between insulin and fat mass percentage are described at the age of 3 months [9], we did not find associations between insulin, glucose, and HOMA-IR at 4 months and body composition during the first year of life; these associations were seen at 2 years of age. This may suggest that the reduction in insulin sensitivity early in life affects body composition at a later age rather than the opposite. There is conflicting evidence comparing IGF-1 levels in formula-fed infants with different protein intakes [10,11]. In contrast to the studies described above, we found no significant differences in IGF-1 level between formula-fed and breast-fed infants. Furthermore, no associations between IGF-1 at the age of 4 months and anthropometric parameters and body composition were found until the age of 2 years, which is in contrast to other studies in newborn infants [11,29,30]. In contrast to our findings, both the CHOP and EPOCH found significantly higher IGF-BP2 levels in their low protein group compared to their high protein group [10,11]. Furthermore, their breast-fed groups showed higher IGF-BP2 levels compared to both formula groups. In line with our results, IGF-BP3 was not different between the formula groups. Both studies found significantly lower levels of IGF-BP3 in their breast-fed group than in their formula-fed groups. IGF-BP1 was not determined in either study. These different findings highlight the complexity of the effect of IGF-1 on growth and body composition and the role of protein intake in infants: Both protein quantity and quality may be relevant to consider in this respect.

We found that leptin levels were not affected by protein intake. Although leptin levels at four months of age did not differ across the three feeding groups, they were positively associated with anthropometric parameters and body composition including fat mass, fat free mass, and fat mass percentage through the age of 2 years. There is conflicting evidence regarding leptin levels of formula-fed and breast-fed infants [31,32]. Our findings are in contrast with a study in healthy term-born infants (*n* = 197) investigating the fasting serum levels of appetite-regulating hormones including leptin. They found that formula-fed infants had significantly higher leptin levels at 3 months of age compared to breast-fed infants. In line with our results, these values correlated positively with fat mass percentage at 3 months and 6 months [9]. Our study has several strengths and limitations. We used a unique customized blend of essential amino acids in our mLP formula. Its composition is based on outcomes of clinical trials conducted in healthy term-born formula-fed infants. Furthermore, at the age of 4 months, no complementary feeding was introduced, so our findings could be attributed solely to a difference in protein intake by infant formula used at that age. An important limitation was that the infants were about one month of age at enrollment, and more than 50% of the infants in the formula groups received breast-milk for some period, with unknown effect on the outcomes studied. The low proportion of participants whose blood was sampled, was partly caused by unsuccessful blood draws. For practical reasons related to the type of measurement only limited numbers of participants had a successful body composition measurement at the ages of 1 and 2 years.

In conclusion, at the age of 4 months, infants fed a mLP formula did not differ from those fed a CTRL formula in their metabolic profile despite a 20%-reduction in protein intake, which emphasizes the safety of this low-protein infant formula. We found that formula-fed infants had lower insulin sensitivity compared to breast-fed infants. Furthermore, long-lasting associations were found between all markers and growth and body composition except for IGF-1. Our results showed that the differences found in growth and body composition between formula-fed and breast-fed infants [23] cannot be explained by differences in IGF-1 level at four months of age, but do suggest that the binding proteins and insulin sensitivity play an important role in growth and body composition trajectories.

## Figures and Tables

**Figure 1 nutrients-13-01159-f001:**
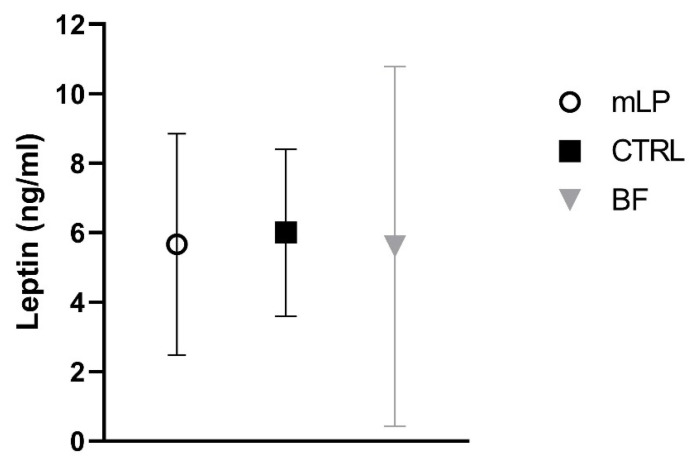
Leptin at 4 months of age. Mean ± SD.

**Figure 2 nutrients-13-01159-f002:**
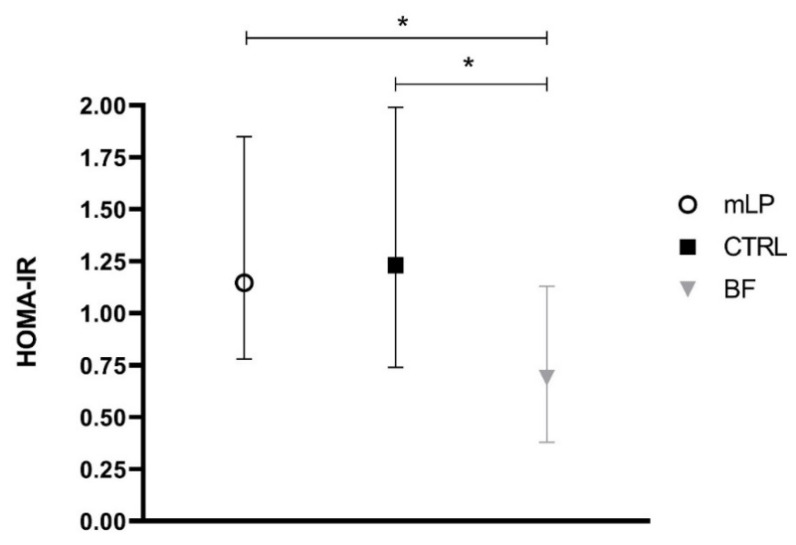
HOMA-IR at 4 months of age. Median (IQR), * *p* < 0.001.

**Table 1 nutrients-13-01159-t001:** Infant characteristics.

	mLP (*n* = 53)	CTRL (*n* = 45)	BF (*n* = 36)	Total Group (*n* = 134)
Infant characteristics				
Boys [*n* (%)]	23 (43.4)	20 (44.4)	14 (38.9)	57 (42.5)
Caucasian [*n* (%)]	45 (84.9)	37 (82.2)	31 (86.1)	113 (84.3)
Gestational age (Wk)	39.5 ± 1.2	39.8 ± 1.2	39.9 ± 1.1	39.7 ± 1.2
Birth weight (g)	3418 ± 366	3402 ± 437	3400 ± 284	3408 ± 370
Caesarean delivery [*n* (%)]	18 (34.0)	14 (31.1)	7 (19.4)	39 (29.1)
Ever breast-fed [*n* (%)]	27 (50.9)	20 (44.4)	36 (100)	83 (61.9)
Age at baseline (d)	30 ± 9	31 ± 10	31 ± 8	31 ± 9
Age at visit 4 mo (mo)	3.9 ± 0.1	3.9 ± 0.1	3.9 ± 0.1	3.9 ± 0.1
Randomized in Amsterdam	35 (66.0)	25 (55.6)	27 (75)	87 (64.9)

Data are presented as mean ± SD or *n* (%). BF, breast-fed; CTRL, control formula; mLP, modified low-protein formula.

**Table 2 nutrients-13-01159-t002:** Anthropometric parameters at 4 months, 6 months, 1 year, and 2 years of age.

	mLP (*n* = 53)		CTRL (*n* = 45)		BF (*n* = 36)	
	*n*		*n*		*n*	
Anthropometry						
Weight (g)						
4 months	53	6647 ± 648	45	6621 ± 644	36	6307 ± 774
6 months	53	7695 ± 804	44	7770 ± 753	36	7241 ± 827
1 year	44	9548 ± 1143	37	9621 ± 1036	29	9302 ± 860
2 years	41	12,400 ± 1598	32	12,195±1200	28	12,103 ± 1040
Length (cm)						
4 months	53	63.9 ± 2.0	45	63.8 ± 2.1	36	63.4 ± 2.3
6 months	53	68.0 ± 2.2	44	68.3 ± 2.4	36	67.0 ± 2.5
1 year	44	75.7 ± 2.6	37	75.8 ± 2.4	29	74.7 ± 2.2
2 years	41	86.9 ± 3.4	32	86.8±3.1	28	86.1 ± 2.3
Body composition						
Fat mass (g)						
4 months	52	1738 ± 335	43	1712 ± 414	33	1586 ± 461
6 months	50	1985 ± 430	39	2023 ± 477	36	1861 ± 461
1 year	15	1916 ± 950	12	1941 ± 701	16	1805 ± 822
2 years	24	2917 ± 1285	14	2156 ± 1123	22	2427 ± 874
Fat (%)						
4 months	52	25.7 ± 3.4	43	25.3 ± 4.7	33	24.5 ± 5.0
6 months	50	25.4 ± 3.6	39	25.7 ± 4.5	36	25.2 ± 4.6
1 year	15	19.1 ± 8.5	12	20.1 ± 6.6	16	19.1 ± 8.0
2 years	24	22.2 ± 7.7	14	17.1 ± 7.7	22	19.8 ± 7.1
Fat free mass (g)						
4 months	52	5014 ± 463	43	5018 ± 471	33	4796 ± 484
6 months	50	5789 ± 549	39	5781 ± 559	36	5455 ± 580
1 year	15	7921 ± 1123	12	7621 ± 787	16	7453 ± 801
2 years	24	9858 ± 605	14	10,099 ± 627	22	9858 ± 1148

Data are presented as mean ± SD. BF, breast-fed; CTRL, control formula; mLP, modified low-protein formula.

**Table 3 nutrients-13-01159-t003:** Metabolic and hormonal markers at 4 months of age.

	mLP		CTRL		BF	
	*n*		*n*		*n*	
Glucose (mmol/L)	53	4.95 ± 0.38	44	4.87 ± 0.34	36	4.88 ± 0.37
Insulin (µU/mL)	52	5.35 (3.65, 8.08)	44	5.60 (3.45, 9.28)	36	3.15 (1.85, 5.28)
HOMA-IR	52	1.15 (0.78, 1.85)	44	1.23 (0.74, 1.99)	36	0.69 (0.38, 1.13)
Leptin (ng/mL)	45	5.66 ± 3.19	38	6.00 ± 2.40	31	5.60 ± 5.18
IGF-1 (ng/mL)	48	25.41 (6.23, 88.13)	39	27.66 (8.78, 81.36)	32	36.29 (5.19, 70.18)
IGF-BP1 (ng/mL)	47	7.95 (4.70, 14.42)	39	8.37 (6.14, 11.98)	32	15.60 (7.90, 20.62)
IGF-BP2 (ng/mL)	48	523 ± 230	38	471 ± 154	32	594 ± 179
IGF-BP3 (ng/mL)	48	3050 ± 612	39	3002 ± 521	32	2777 ± 548

Data are presented as mean ± SD or median (IQR). BF, breast-fed; CTRL, control formula; mLP, modified low-protein formula.

**Table 4 nutrients-13-01159-t004:** Differences in metabolic and hormonal markers between feeding groups at 4 months of age.

	mLP vs. CTRL (mLP Minus CTRL)			mLP vs. BF (mLP Minus BF)			CTRL vs. BF (CTRL Minus BF)		
		95% CI	*p*		95% CI	*p*		95% CI	*p*
Glucose ^1^ (mmol/L)	0.07	−0.07, 0.22	0.33	0.06	−0.1, 0.21	0.45	−0.01	−0.17, 0.15	0.87
Insulin ^2^ (µU/mL)	0.96	0.74, 1.23	0.72	1.69	1.30, 2.20	<0.001	1.77	1.34, 2.33	<0.001
HOMA-IR ^2^	0.96	0.74, 1.25	0.79	1.70	1.29, 2.25	<0.001	1.76	1.32, 2.35	<0.001
Leptin ^1^ (ng/mL)	−0.44	−2.00, 1.12	0.58	0.05	−1.60, 1.70	0.95	0.49	−1.23, 2.20	0.57
IGF-1 ^2,3^ (ng/mL)	0.86	0.30, 2.49	0.86	0.92	0.30, 2.82	0.72	0.79	0.24, 2.56	0.83
IGF-BP1 ^2^ (ng/mL)	0.92	0.67, 1.27	0.63	0.58	0.41, 0.81	<0.01	0.62	0.44, 0.89	<0.01
IGF-BP2 ^1^ (ng/mL)	46	−37, 129	0.27	−69	−156, 17	0.12	−115	−207, −24	0.014
IGF-BP3 ^1^ (ng/mL)	36	−204, 276	0.77	277	24, 531	0.03	241	−24, 507	0.08

Values are differences between feeding groups compared with linear regression analysis adjusted for sex. ^1^ differences between the groups; ^2^ differences between the groups in ratio; ^3^
*p* value calculated with Mann-Whitey U. BF, breast-fed; CTRL, control formula; mLP, modified low-protein formula.

**Table 5 nutrients-13-01159-t005:** Metabolic and hormonal markers at 4 months in relation to anthropometric parameters and body composition through the age of 2 years.

Metabolic and Hormonal Markers	Age	Weight, g			Length, cm			Fat Mass, g			Fat Free Mass, g			Fat Mass, %		
		*B*	95%CI	*p*	*B*	95%CI	*p*	*B*	95%CI	*p*	*B*	95%CI	*p*	*B*	95%CI	*p*
Glucose at 4 months																
	4 months	188.88	−232.06, 609.83	0.38	−0.70	−1.14, 1.00	0.90	156.33	−155.94, 468.60	0.33	37.80	−227.88, 303.47	0.78	1.69	−0.93, 4.31	0.21
	6 months	228.74	−192.24, 649.71	0.29	0.57	−0.40, 1.64	0.29	91.89	−222.39, 406.18	0.57	175.87	−91.56, 443.31	0.20	0.30	−2.34, 2.93	0.83
	1 year	23.27	−406.30, 452.83	0.92	0.83	−0.26, 1.91	0.14	−24.64	−483.45, 434.16	0.92	256.08	−136.87, 649.03	0.20	−0.62	−4.48, 3.24	0.75
	2 years	338.33	−105.30, 781.96	0.13	0.99	−0.13, 2.12	0.08	−97.89	−488.58, 292.79	0.62	337.85	4.01, 671.68	0.047	−1.13	−4.42, 2.15	0.50
Insulin at 4 months																
	4 months	9.08	−23.32, 41.49	0.58	−0.01	−0.09, 0.07	0.86	9.84	−13.96, 33.65	0.42	−4.13	−24.49, 16.23	0.69	0.14	−0.05, 0.34	0.15
	6 months	20.17	−12.28, 52.62	0.22	0.04	−0.04, 0.12	0.32	16.90	−6.91, 40.71	0.16	4.20	−16.17, 24.57	0.69	0.16	−0.04, 0.36	0.12
	1 year	8.66	−24.83, 42.16	0.61	0.05	−0.04, 0.13	0.26	5.85	−51.18, 62.88	0.84	−8.02	−57.19, 41.15	0.75	0.09	−0.39, 0.57	0.71
	2 years	17.55	−16.88, 51.97	0.32	0.08	−0.01, 0.16	0.09	25.20	−3.67, 54.06	0.09	−30.84	−55.61, −6.08	0.01	0.23	−0.01, 0.48	0.06
HOMA-IR at 4 months																
	4 months	40.24	−99.10, 179.59	0.57	−0.04	−0.39, 0.31	0.82	42.07	−60.31, 144.45	0.42	−14.47	−102.07, 73.13	0.75	0.61	−0.25, 1.46	0.16
	6 months	86.10	−53.41, 225.61	0.23	0.18	−0.17, 0.54	0.30	70.91	−31.56, 173.38	0.17	20.56	−67.12, 108.24	0.64	0.65	−0.20, 1.51	0.13
	1 year	27.70	−115.90, 171.29	0.70	0.21	−0.15, 0.57	0.25	23.63	−221.14, 268.40	0.85	−29.88	−241.04, 181.28	0.78	0.37	−1.69, 2.44	0.72
	2 years	71.72	−74.79, 218.22	0.34	0.32	−0.05, 0.69	0.09	96.83	−23.49, 217.15	0.11	−119.97	−223.18, −16.75	0.02	0.92	−0.09, 1.92	0.08
Leptin at 4 months																
	4 months	75.05	30.55, 119.55	0.001	0.12	0.01, 0.22	0.04	40.77	6.92, 74.62	0.02	33.25	5.31, 61.19	0.02	0.31	0.03, 0.58	0.03
	6 months	76.37	31.86, 120.88	<0.010	0.13	0.02, 0.24	0.02	47.07	13.33, 80.70	<0.01	31.34	3.54, 59.14	0.03	0.33	0.06, 0.61	0.02
	1 year	35.79	−9.16, 80.74	0.12	0.13	0.02, 0.24	0.02	84.60	4.78, 164.42	0.04	−52.73	−118.65, 13.19	0.12	0.87	0.22, 1.53	<0.015
	2 years	63.11	17.35, 108.87	<0.01	0.12	0.003, 0.23	0.04	53.34	13.06, 93.61	<0.01	22.23	−11.02, 55.47	0.19	0.29	−0.04, 0.62	0.09
IGF-1 at 4 months																
	4 months	1.84	−2.02, 5.70	0.35	0.001	−0.01, 0.01	0.80	1.05	−1.90, 4.01	0.48	0.68	−1.83, 3.19	0.59	0.01	−0.02, 0.03	0.47
	6 months	1.05	−2.82, 4.92	0.59	0.000	−0.01, 0.01	0.98	−0.07	−3.03, 2.89	0.96	1.19	−1.32, 3.71	0.35	−0.01	−0.03, 0.02	0.69
	1 year	−2.85	−6.89, 1.20	0.17	−0.003	−0.01, 0.01	0.5	−2.16	−7.14, 2.83	0.40	3.24	−0.99, 7.46	0.13	−0.02	−0.07, 0.02	0.25
	2 years	−1.84	−6.04, 2.36	0.39	−0.007	−0.02, 0.00	0.16	2.05	−2.05, 6.14	0.33	0.14	−3.34, 3.61	0.94	0.01	−0.02, 0.05	0.39
IGF-BP1 at 4 months																
	4 months	−8.36	−17.97, 1.25	0.09	−0.033	−0.06, −0.01	0.01	−3.27	−10.58, 4.04	0.38	−4.67	−10.63, 1.29	0.12	−0.02	−0.08, 0.05	0.61
	6 months	−12.21	−21.82, −2.59	0.01	−0.046	−0.07, −0.02	<0.001	−2.23	−9.52, 5.07	0.55	−9.83	−15.78, −3.88	0.001	0.02	−0.04, 0.08	0.56
	1 year	−18.15	−27.84, −8.47	<0.001	−0.051	−0.08, −0.03	<0.001	−1.51	−9.25, 6.23	0.70	−16.12	−22.45, −9.80	<0.001	0.03	−0.04, 0.09	0.41
	2 years	−19.09	−28.79, −9.38	<0.001	−0.042	−0.07, −0.02	<0.001	−15.51	−25.07, −5.96	0.001	−1.50	−9.36, 6.35	0.71	−0.11	−0.19, −0.03	<0.01
IGF-BP2 at 4 months																
	4 months	−0.86	−1.66, −0.06	0.04	−0.002	−0.003, 0.00	0.06	−0.28	−0.90, 0.35	0.39	−0.55	−1.06, −0.04	0.04	−0.001	−0.01, 0.00	0.84
	6 months	−1.33	−2.13, −0.53	0.001	−0.004	−0.01, −0.00	<0.001	−0.40	−1.03, 0.23	0.21	0.88	−1.40, −0.37	<0.001	−0.001	−0.01, 0.00	0.77
	1 year	−1.07	−1.89, −0.25	0.01	−0.003	−0.01, −0.00	<0.001	−0.08	−0.99, 0.83	0.86	−1.48	−2.22, −0.73	<0.001	0.002	−0.01, 0.01	0.56
	2 years	−1.22	−2.06, −0.39	<0.01	−0.006	−0.01, −0.00	<0.001	−0.08	−0.92, 0.77	0.86	−0.44	−1.13, 0.25	0.21	0.001	−0.01, 0.01	0.68
IGF−BP3 at 4 months																
	4 months	0.41	0.13, 0.69	<0.01	0.001	−0.0002, 0.00	0.13	0.21	−0.01, 0.42	0.06	0.19	0.01, 0.37	0.04	0.001	−0.000130, 0.003448	0.07
	6 months	0.55	0.27, 0.82	<0.001	0.001	−0.00002, 0.00	0.06	0.24	0.03, 0.45	0.03	0.31	0.13, 0.49	<0.001	0.001	−0.001, 0.003	0.15
	1 year	0.44	0.15, 0.72	<0.01	0.001	−0.0001, 0.00	0.08	0.14	−0.16, 0.45	0.4	0.18	−0.08, 0.44	0.17	0.001	−0.001, 0.003	0.62
	2 years	0.27	−0.03, 0.56	0.08	0.000	−0.0005, 0.00	0.46	−0.27	−0.56, 0.02	0.07	0.28	0.03, 0.53	0.03	−0.002	−0.005, 0.0004	0.10

Values are associations between metabolic and hormonal markers and growth and body composition using linear mixed model analysis adjusted for feeding group and sex. B, coefficients; BF, breast-fed; CTRL, control formula; mLP, modified low-protein formula.

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
