# Peer review of "Early-Life Metabolic and Hormonal Markers in Blood and Growth until Age 2 Years: Results from a Randomized Controlled Trial in Healthy Infants Fed a Modified Low-Protein Infant Formula"

_nutrients, 2021, doi:10.3390/nu13041159_

Round 1

Reviewer 1 Report

This study adds to the knowledge about infant feeding, metabolic and hormonal markers associated with infant feeding and child anthropometrics and is absolutely worth reporting.

However, I have several suggestions that I think could make the paper more reader-friendly and make it scientifically more valid by following reporting guidelines (Authors may want to check STROBE, CONSORT).

A general comment: This study is a combination of a RCT and plenty of analyses that are not based on randomization. From scientific point of view, I would consider the RCT part most valuable, and the analyses comparing markers with non-randomized breastfeeding group and longitudinal comparison between biological markers and anthropometric variables, secondary.

Related to this, authors write “A breast-fed (BF) group served as reference.” (abstract, methods) This might be a language thing, but to me this sounds like the main comparisons are those to the made between BF group and others. If authors agree, that the RCT is valuable, please consider re-formulating this. Eg, in addition, comparisons with BF group were done.

Introduction:

  • It would be reader-friendly to mention in the introduction that the intervention groups did not differ in weight/anthropometric variables (by age XX) in the published study (ref 24) .

Methods:

  • It would be reader-friendly to include subtitle “intervention groups” and in addition to current information mention the age of infants at randomization (apparently 1 month), mention at least whether formulas were isocaloric and how this was achieved, how long was the intervention duration, and whether amounts of consumption were pre-determided. This information has been apparently reported before, but is crucial to understand while reading this paper.

  • Randomization: In which group were the partly-breastfed children allocated? Did the breastfeeding group include only children who were exclusively breastfed at the randomization? If so, this should be mentioned.

  • Statistical methods: In the linear regression model, were all 3 groups (two intervention groups and the breastfeeding group) included in the same model or were three models with two groups in each run separately? Please report, which way was used. As the breastfeeding group was not randomized, I would keep it separate from the analyses comparing the intervention groups. If you have included all groups in the same model, please describe in your response, whether findings would be the same when excluding the breastfeeding group from the analyses and consider reporting such results.

Results

  • Please mention something about compliance and amount of formula consumed in the formula groups. Where there other sources of protein in the diet, such as breastmilk and solid foods (at what ages this was assessed)?
  • Please report the protein intake in intervention groups (if available, at 4 months would be especially interesting). It has been mentioned in the introduction that the groups differed in protein intake, but only at abstract and discussion that they differed by 20%. It is also unclear, at which age this difference was observed, and was it based on total diet. I addition, number of children included in this study seem to differ from that reported previously, and therefore the protein (and energy) intakes, and whether they are based on assumed formula intake, actual formula intake only or whole diet, should be reported.
  • “Of these samples”-sentence: it takes a while to understand whether referring to children with or without blood samples. Please consider revising.
  • Were the comparison done by intention to treat (if neglecting the missing blood samples) or per protocol? Consider including a flow-chart.
  • Table 5 Title: “makers” should maybe be “markers”. Makers appears also elsewhere in the text.
  • Consider revising Table 5 output or title. A quick reader would assume that glucose has been assessed at 4, 6 months, 1 & 2 years, and Weight just once. Tile could be e.g. Metabolic and hormonal markers at 4 months in relation to anthropometric parameters and body composition at ages… The same applies to the supplementary table. Consider reporting the RCT separately from the non-randomized associations.
  • Results page “At the age of 2 years, glucose was positively associated with fat free mass.” Do you mean: “Glucose at 4 months was positively associated with fat free mass at 2 years”? If so, please consider clarifying all similar sentences. (a quick reader may not remember data collection points)

Discussion

  • Please, include discussion about weaknesses of the study, and generalizability. (reporting guidelines suggest this)
  • “These different findings might be caused by the difference in total protein intake between the intervention- and the control group included in these studies.“ Was the difference in protein intake between intervention groups smaller in present study than in previous studies? Could the not-expected results (no difference in metabolic markers) be due to only a small difference in protein intake between groups? Please, consider discussing.
  • Please, consider discussing what could explain the differences between the breastfed and formula fed children (are there other explanations than protein? How much protein breastmilk contains?) As the breasfed group was not randomized, could there be potential confounding factors? I noticed that child sex that was included in the model.
  • The Concluding paragraph includes results apparently from another paper under review. I don’t think that the concluding paragraph of this study should include such information but focus on concluding findings from the present paper and their application. There are apparently some unfinished sentences in the conclusions-chapter, so it is a bit difficult to understand.

Other

  • If you want to shorten your paper, I think that reporting mean ages at visits 6 mo, 1, 2 years is not necessary and could be left out from table 1.

Author Response

Dear sir/madam, please find our response to the reviewer attached. 

Author Response

(The authors gave the same response as above.)
